# Is Cemented Dual-Mobility Cup a Reliable Option in Primary and Revision Total Hip Arthroplasty: A Systematic Review

**DOI:** 10.3390/jpm13010081

**Published:** 2022-12-29

**Authors:** Gianluca Ciolli, Guillaume Mesnard, Etienne Deroche, Stanislas Gunst, Cécile Batailler, Elvire Servien, Sébastien Lustig

**Affiliations:** 1Orthopaedic Department, Lyon North University Hospital, Hôpital de La Croix Rousse, Hospices Civils de Lyon, 103 Grande Rue de la Croix Rousse, 69004 Lyon, France; 2Orthopaedics and Traumatology, Fondazione Policlinico Universitario A. Gemelli IRCCS, Sacred Heart Catholic University, 00168 Rome, Italy; 3Univ Lyon, Claude Bernard Lyon 1 University, IFST-TAR, LBMC UMR_T9406, 69622 Lyon, France; 4LIBM—EA 7424, Interuniversity Laboratory of Biology of Mobility, Claude Bernard Lyon 1 University, 69622 Lyon, France

**Keywords:** hip arthroplasty, dual mobility cup, cemented dual mobility, dislocation, instability, hip revision

## Abstract

**Background**: Instability is a common complication following total hip arthroplasty (THA). The dual mobility cup (DMC) allows a reduction in the dislocation rate. The goal of this systematic review was to clarify the different uses and outcomes according to the indications of the cemented DMC (C-DMC). **Methods**: A systematic review was performed using the keywords “Cemented Dual Mobility Cup” or “Cemented Tripolar Cup” without a publication year limit. Of the 465 studies identified, only 56 were eligible for the study. **Results**: The overall number of C-DMC was 3452 in 3426 patients. The mean follow-up was 45.9 months (range 12–98.4). In most of the cases (74.5%) C-DMC was used in a revision setting. In 57.5% DMC was cemented directly into the bone, in 39.6% into an acetabular reinforcement and in 3.2% into a pre-existing cup. The overall dislocation rate was 2.9%. The most frequent postoperative complications were periprosthetic infections (2%); aseptic loosening (1.1%) and mechanical failure (0.5%). The overall revision rate was 4.4%. The average survival rate of C-DMC at the last follow-up was 93.5%. **Conclusions**: C-DMC represents an effective treatment option to limit the risk of dislocations and complications for both primary and revision surgery. C-DMC has good clinical outcomes and a low complication rate.

## 1. Introduction

The use of the dual mobility cup (DMC) is an established practice in hip replacement surgery, which could ensure higher implant stability, physiological mobility of the hip joint and reduce wear. DMC is considered one of the major current strategies to prevent and treat hip instability, which is the first reason for total hip arthroplasty (THA) revision. The excellent long-term results of DMC justify their steady increase in recent years, both in primary complex cases and in recurrent hip instability after THA [1,2,3,4,5].

DMC can be used as the first implant surgery in complex cases of osteoarthritis (OA), including obese patients or those with neuromuscular or neurological diseases; and in fractures, such as acetabular fractures, femoral neck fractures (FNF), or pathological fractures [6,7,8,9]. Another common use of DMC is in revision surgery of total or partial hip arthroplasties or following the failure of a previous osteosynthesis [10,11].

In complex cases, including bone defects, DMC can be cemented directly into the bone or acetabular brace, such as primary or revision surgery [12,13]. While many reviews in the literature investigate and describe DMC [14,15], there is no specific analysis of cemented DMC (C-DMC).

The study aims to provide a systematic review concerning the C-DMC and a specific analysis of its application, in terms of complications, clinical results, survivorship rate, and radiographic findings. Secondary objectives are to describe the outcomes of primary and revision surgery and with or without acetabular reinforcement.

## 2. Materials and Methods

### 2.1. Search Strategy and Design

A systematic literature review was performed following the 2009 PRISMA (Preferred Reporting Items for Systematic Reviews and Meta-analyses) guidelines. The following online electronic databases were used: Cochrane Database of Systematic Reviews, PubMed, EMBASE and Google Scholar. The search strategy had no data limit and was conducted until 31 December 2021. The following keywords, and their MeSH terms in any possible combination, were used: “Cemented Dual Mobility Cup” OR “Cemented Tripolar Cup”. A minimum mean follow-up of 1 year was considered to allow the evaluation of all early complications and outcomes.

### 2.2. Eligibility Criteria

Inclusion criteria were (1) patients undergoing C-DMC for any reason; (2) all levels of evidence; (3) full-text articles in indexed journals; (4) studies in English; (4) human studies; (5) mean follow-up of at least 12 months. Exclusion criteria were (1) nonoperative studies; (2) studies using different types of cups; (3) studies in which C-DMC was been used in less than 10% of cases; (4) studies in which DMC data were not specifically described; (5) reviews; (6) surveys or case reports; (7) book chapters; (8) congress abstract; (9) surgical technical reports; (10) expert opinions or letters to the editor; (11) cadaveric or animal studies.

### 2.3. Study Selection

The study selection was conducted by two independent reviewers (C.G. and M.G.). Articles were identified based on the title and abstract. If necessary, full-text articles were obtained to screen. After excluding the unacceptable studies, the full texts of the remaining studies were assessed. Any disagreements between reviewers were resolved through discussion with a third author (D.E.). Finally, the included articles’ references have been evaluated to highlight further relevant items useful for the search. In the case of multiple studies involving the same case series with different follow-ups, only the manuscript with the longest follow-up was selected.

### 2.4. Data Extraction and Analysis

A standard data extraction form was used which included the following: (1) study details: author, year, country, study design, level of evidence; (2) study population: cohort, population size, gender, age at the time of surgery, body mass index (BMI); (3) follow-up information: patients at follow-up, mean follow-up, patients lost to follow-up; (4) surgical approach to the hip; (5) C-DMC information: number, implant, cement, graft, primary or revision procedure, “cup-in-cup” technique, acetabular reinforcement; (6) outcomes: preoperative and postoperative Harris Hip Score (HSS), postoperative Postel Merle d’Aubigné (PMA); (7) postoperative complications: mechanical failures, aseptic loosening, infections, dislocations, intra-prosthetic dislocations (IPD); (8) radiographic complications: peri-acetabular radiolucent lines (RLLs), Brooker heterotopic ossification (HO); (9) survivorship rate. In the presence of comparative studies, in which the C-DMC are compared with different treatment techniques, only the C-DMC data were obtained for the study; while in the presence of overall values with the other different cups, these values were not taken into consideration for the statistics.

### 2.5. Methodological Quality Assessment

The study quality of all included studies was evaluated using the MINORS (Methodological Index for Non-Randomized Studies) criteria. Each item was scored from 0 to 2, with maximum scores of 16 for non-comparative studies and 24 for comparative studies. Each study included was scored by 2 authors (C.G. and M.G.).

### 2.6. Statistical Analysis

The kappa (k) value was used to evaluate consensus among reviewers in item selection. The agreement was classified as poor when k < 0.30, partial 0.30 < k < 0.60 and total with k > 0.60. Given the high heterogeneity among the studies, a meta-analysis was not performed; however, indirect comparisons were made.

## 3. Results

### 3.1. Literature Search and Study Characteristics

The initial search found 465 studies. After removing the duplicates, the remaining items were screened based on the title and abstract. The full texts of the remaining 86 articles were examined. Finally, according to the eligibility criteria, 56 articles were included in the systematic review [12,14,15,16,17,18,19,20,21,22,23,24,25,26,27,28,29,30,31,32,33,34,35,36,37,38,39,40,41,42,43,44,45,46,47,48,49,50,51,52,53,54,55,56,57,58,59,60,61,62,63,64,65,66,67,68] (Figure 1).

Among the reviewers, there was excellent agreement involving the title (k = 0.90; 95% CI, 0.88 to 0.92), the abstract (k = 0.91; 95% CI, 0.89 to 0.93), and the full text (k = 0.93; 95% CI, 0.92 to 0.94). Of the selected articles published between 2008 and 2021, ten reports are comparative studies (17.9%).

Most of the studies were retrospective (n = 44, 78.6%), while the remaining were prospective (n = 12, 21.4%). Forty-two (75%) studies were level IV evidence, 12 (21.4%) studies were level III, and two (3.6%) studies were level I.

The mean MINORS score was 11 (range 10–14) and 20 (range 17–24), for the non-comparative and comparative studies, respectively.

Most of the studies were conducted in France (28.1%), Egypt and the United States (both 10.7%) (Table 1).

### 3.2. Overall Demographic Data

The overall number of patients was 4675 (4701 hips). Finally, 3452 C-DMC in 3426 patients were found. The patients who reached the last follow-up and who were evaluated for clinical and radiographic outcomes are 3162 (92.3%).

Females are more represented (62.9%) than males (37.1%). The mean age at the time of surgery was of 71.5 years (mean range 67–82), only one study doesn’t have the mean age value. The mean follow-up was 45.9 months (range 12–98.4). The mean BMI was 26.8 kg/m^2^ (range 24–29.7) (Table 1, Table 2 and Table 3).

### 3.3. Operative Technique and Implants

Overall, 3452 C-DMCs are included. The posterolateral approach was the most used surgical approach (75.3%). The anterior approach was used in 14% of cases, the lateral approach in 5.3% of cases, and an extensive approach was used in 5.4% of cases. The most used implant was the Avantage dual mobility cup (Zimmer Biomet©, Warsaw, IN, USA), used in 17 studies and 54.2% of all cases, followed by the DM Novae cemented cup (Serf©, Décines-Charpieux, France) (7 studies, 14.2%) (Table 2 and Table 3).

### 3.4. Complications and Cup Survival Rate

The overall dislocation rate from 51 studies was 3.1%. The systematic review reported 7 cases of IPD, in four different studies. The overall IPD rate is 0.2%. The most frequent postoperative complications were periprosthetic infections (3%); aseptic loosening (1.3%) and mechanical failure (0.5%). A total of 138 revisions are reported (revision rate of 4%), in 48 studies reporting data eligible for the study. The average survival rate of the DMC at the last follow-up was evaluated in 43 studies (76.8%). At 45.9 months follow-up, the overall survival rate was 93.5% (range 83.1–100) (Table 4).

### 3.5. Radiographic Outcomes

Among the systematic review, 36 studies (64.3%) showed radiological information about peri-acetabular radiolucent lines (RLLs), and only 26 studies (46.4%) investigated Brooker HO. The rate of periacetabular RLLs was 3.2%, while the HO rate was 6.7% (Table 5).

### 3.6. Functional Outcomes

The preoperative level of function was assessed with the HHS in 22 studies (59.3%). The mean preoperative HHS value was on average 43. The mean postoperative HHS value, evaluated in 34 studies (60.7%), was 76.7 (Table 5). The mean pre-operative PMA was 10.4 in 10 studies (17.9%). Instead, the post-operative PMA was 14.7, in 14 studies (25%) (Table 5).

### 3.7. Cemented Dual Mobility Cups in the Primary Setting

C-DMCs were used in 25.5% of cases as the primary setting, in 19 different studies (33.9%). In 40.5% of those cases, C-DMCs were used for the treatment of FNF, in 27.9% of cases as a treatment of OA complex cases, in 25.2% of cases as a treatment of pathological lesions (all cases of peri-acetabular metastasis cases except one case of pathological femur fracture) [59].

Finally, in 6.4% of the cases, a C-DMC was used in the treatment of acetabular fractures (Table 6).

### 3.8. Cemented Dual Mobility Cups in the Revision Setting

In 74.5% of cases, a C-DMC was used in the revision setting, such as the revision of previous THA or fixation failures. The use of C-DMC in revision cases was described in 42 studies (75%) (Table 7).

### 3.9. Acetabular Reinforcement

In most cases (57.5%), the C-DMCs were used without acetabular reinforcement, cementing the acetabular component directly on the bone resulting in a low loosening rate and good cup survivorship, between 93 and 100% at the last followup (Table 8).

In 39.6% of the total cases, in 30 studies, a DMC was cemented into an acetabular reinforcement.

The most common AR used were the Kerboull cross-plates and Burch–Schneider Cage (Table 9).

### 3.10. Cup-in-Cup Technique

DMC was cemented into a pre-existing cup following the “cup-in-cup” technique, in 106 cases (3.1%) (Table 3). In all cases, this procedure was used as a revision setting after the failure of a previous THA (Table 10).

### 3.11. Acetabular Bone Grafting

In the current study, the use of bone grafting was observed in 22 (39.3%) manuscripts. Overall, acetabular bone grafting has been used in 16.3% of C-DMC. In most cases, it was an allograft (93.2%), while in a minor part it was a synthetic graft (4.9%) or autograft (1.9%).

Acetabular bone grafting was used in 3% of the primary C-DMC setting and in 19% of the revision C-DMC setting. In cases where an acetabular reinforcement was utilized, acetabular bone grafting was used in 24.2% of cases, while only in 9.8% of cases in the absence of acetabular reinforcement (Table 2).

## 4. Discussion

This review allowed us to appreciate the satisfactory results of C-DMC. Dislocation rates found are low and the survivorship of the implants in both primary and revision surgery is satisfactory. Clinically, the improvement of the functional outcomes is significant, and radiologically, RLLs are rare.

The clinical and radiographic outcomes of cementless DMCs are excellent at medium and long follow-ups with an overall survivorship of 95% at 10-year follow-ups and demonstrated low revision and dislocation rates, without major polyethylene wear concerns [16,68,69,70,71,72,73,74,75,76,77,78,79,80].

### 4.1. Cemented Dual Mobility Cup in the Setting of Primary Surgery

In the present study, was observed the use of C-DMC in a primary setting in about a quarter of the surgeries (25.5%). C-DMC is an established treatment of hip OA, especially in association with other comorbidities that could increase the risk of dislocation and loosening, such as acetabular deficiency, abductor deficiency, prior acetabular fracture fixation or high-demanding physical activity patients [32].

Tabori-Jensen et al. [61] in a randomized controlled study in elderly osteoarthrosis patients with 2 years’ follow-up comparing cementless DMC with C-DMC have demonstrated that cemented fixation of the Avantage DMC seems safer in elderly patients, with less implant migration.

Moreover, C-DMC represents an option in the management of FNF. In those cases, Tarasevicius et al. [15] reported a 100% survival rate one year after treatment of 42 patients with C-DMC. Lamo-Espinosa et al. [67], in a series of 69 elderly (>75 years) and frail patients with a high risk of instability with a median follow-up time of 49.04 months, reported only one case of revision due to aseptic loosening.

Another well-recognized use of C-DMC is in cases of pathological fractures. Lavignac et al. [58] used DMC in the treatment of peri-acetabular metastasis, but although the rate of complications and revisions is low, the mean patient survivorship was 19.5 months due to the progression of the primary disease. Similarly, Wegrzyn et al. [46], in a continuous series of 131 cases of periacetabular metastatic disease treated using a DMC cemented into an acetabular reinforcement device, observed an improvement in the postoperative functional outcome and pain relief, no mechanical failure or aseptic loosening of the acetabular reconstruction and a dislocation rate of 2%.

Finally, in some cases of acetabular fractures, C-DMC could be used. In those cases, it is associated with the combined treatment of fracture reduction and fixation. As reported by Giunta et al. [38], primary C-DMC for acetabular fracture in the elderly population might be a good therapeutic option that allows a return to previous daily life activity. Lannes et al. [57] found that DMC combined with internal fixation, also known as the combined hip procedure (CHP), could be an efficient procedure in selected elderly patients, with a lower level of revision rate compared to the ORIF group alone. In the CHP group, internal fixation was performed before the hip replacement. When the anterior column was involved, a suprapectineal quadrilateral buttress plate was used through a modified Stoppa approach; otherwise, a Kocher-Langenbeck approach was performed with a posterior column or wall osteosynthesis [81,82].

These findings are important and demonstrate that primary surgery with C-DMC is an excellent treatment in complex cases of OA, FNF, acetabular fractures or periacetabular metastases in terms of survival and reduced number of complications.

### 4.2. Cemented Dual Mobility Cup in the Setting of Revision Surgery

As demonstrated by the results of the study, the most frequent use of C-DMC is in revision settings (74.5%).

In cases of failure of a previous THA, C-DMC has shown an excellent role in avoiding further failures and reducing the number of complications. In fact, from the Swedish Hip Arthroplasty Register, Hailer, et al. [18] revealed in a series of 228 patients that DMCs for revision due to instability are associated with a low rate (8%) of re-revisions due to dislocation.

In patients with first-time revision hip arthroplasty due to dislocation, Mohaddes et al. [31] found better short to mid-term implant survival for C-DMC compared with cemented polyethylene cups at four years of follow-up. Stucinskas et al. [44], from the Lithuanian arthroplasty register, observed a significantly lower short-term re-revision rate for dislocations at five-year follow-up, in patients treated with C-DMC (2%) compared to different surgical concepts (9%) when used for first-time hip revisions due to recurrent dislocations.

As emerges from the results of the current review, C-DMC is a valid option in a revision setting with a dislocation rate of 3% and a survival rate of approximately 95% (with a mean follow-up of 49.6 months).

### 4.3. Cemented Dual Mobility Cup into an Acetabular Reinforcement

In the present study, DMC was cemented in an acetabular reinforcement in the revision setting in 83.6%, while in the primary setting only in 16.4% of the cases.

Neri et al. [74] suggested the use of DMC cemented into a reinforcement cage (Kerboull cross-plate or custom reinforcement cage) in cases of acetabular defects 2C or more according to Paprosky’s classification. In a single-center continuous series of 62 patients, Lebeau et al. [30] evidenced that DMC cemented into an acetabular reinforcement at a minimum FU of five years presented good outcomes (PMA 14, HHS score 73) and only 2 cases of dislocation, while Brüggemann et al. [35], in a case series of 69 patients, demonstrated a 4-year survival of 96% and only one case of dislocation.

### 4.4. Cemented Dual Mobility Cup into a Pre-Existing Cup

Among the studies, a minor part (3.2%) of the DMCs was cemented into a pre-existing cup following the surgical technique described by Blumenfeld [83], known as the “Cup-in-cup” technique. This technique, developed for the treatment of acetabular defects, in the current study showed good results in terms of fewer complications and guaranteed a survival rate of 93.6% at a mean follow-up of 35.7 months (Table 10).

### 4.5. Bone Grafting

In the current study, the use of bone grafting was observed in 16.3% of C-DMC. In most cases, bone grafting was used in a revision setting, in combination with acetabular reinforcement, using an allograft. Bone grafting could help the surgeon in cases of loss of acetabular bone stock and unstable components fixation and can be easily combined with cemented fixation of the DMC [84].

### 4.6. C-DMC Complications and Outcomes

In the current review, the overall dislocation rate was 2.9%, and at 45.9 months follow-up, the overall survival rate was 93.5%. These data are similar to the mean rates of dislocation and medium-term survival of single-mobility THA or cementless DMC [25,85,86]. These results must be considered even more due to the use of the C-DMC especially in complex cases or in revision settings.

The overall IPD rate was 0.2%. First described in 2004 by Lecuire [87], intra-prosthetic dislocation (IPD) is a specific complication of DMC. IPD is defined as excessive “wear” of the head-liner interface which induces the separation of the head from the polyethylene, but the present systematic review reported only 7 cases of IPD, showing that this complication is now extremely rare.

Overall, C-DMC demonstrates to be a valid treatment in primary complex cases, such as in revision settings, in terms of survival and reduced number of complications.

The greatest strength of the article is to represent the only systematic review specifically focused on the C-DMC present in the literature. Moreover, over three thousand C-DMCs are considered in the systematic review. Limitations of the study are represented by the low average level of evidence, most of the studies have a level IV of evidence. Furthermore, the nature of the inclusion criteria may have minimally altered the results, due to the wide heterogeneity of the patients included. However, it is assumed that in such a large case series, this possible bias could be minimized.

## 5. Conclusions

The use of C-DMCs is an option used mainly in THA revision surgery, although it also exists in a primary setting. It represents an effective treatment option with good clinical outcomes. The complication rate remains moderate, with a low rate of dislocation in both primary and revision surgery. The loosening and revision rates are also low. These results can be observed whether the implant is cemented directly into the bone or cemented in acetabular reinforcement or a previous well-fixed cup. An acetabular reinforcement was used in almost 40% of cases.”

## Figures and Tables

**Figure 1 jpm-13-00081-f001:**
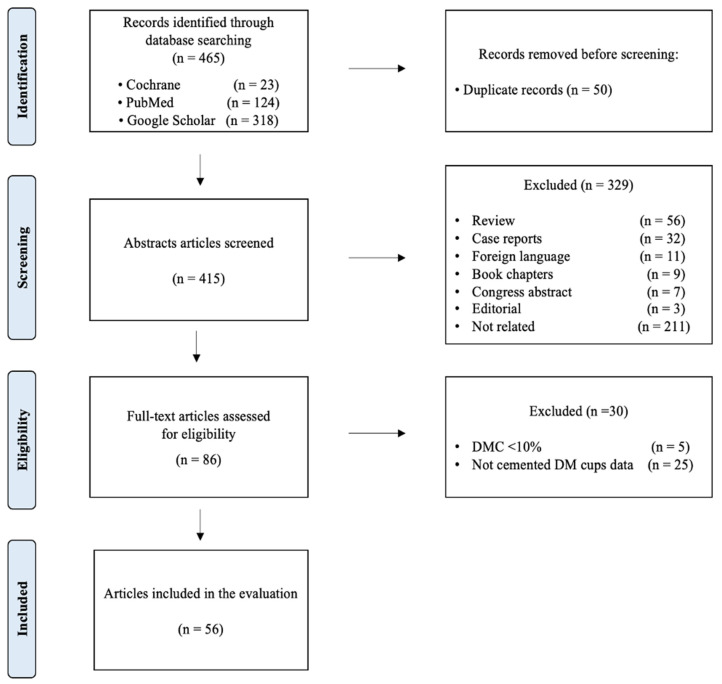
Prisma Flowchart from the identification to the inclusion.

**Table 1 jpm-13-00081-t001:** Characteristics of studies included in the review.

Author	Year	Nationality	Type	Level of Evidence	Number of Patiens	Males	Females	Mean Age (SD) [Range]	BMI (kg/m^2^) (SD) [Range]	Cemented DMC	Patients at Follow-Up	Follow-up (Months) (SD) [Range]	MINORS
Langlais	2008	france	Retrospective	IV	82	/	/	72 [65–86]	/	88	79	36 [24–60]	12
Philippot	2009	france	Retrospective	IV	51	/	/	68.7 [34–92]	/	51	51	60.4 (±17.6) [24–112]	12
Tarasevicius	2010	Sweden	Retrospective comparative	III	42	/	/	75 (±10)		42	42	12	19
Schneider	2011	France	Retrospective	IV	96	25	71	69.9 [34–95]	/	96	77	41 (±29) [1–101]	11
Civinini	2012	Italy	Prospective	IV	24	/	/	69 [51.3–82.4]	/	24	24	39.6 [24–60]	14
Hailer	2012	Sweden	Retrospective	IV	200	/	/	/	/	200	200	24	10
Pattyn	2012	Belgium	Retrospective	IV	36	16	20	70.4 [46–93]	/	37	36	16 [6–27]	11
Mukka	2013	Sweden	Retrospective	IV	34	13	21	75.7 [58–90]	/	34	34	18 [6–36]	11
Toro-Ibarguen	2014	Spain	Retrospective	IV	14	/	/	67.8 [29–90]	/	14	14	64.8	10
Wegrzyn	2014	france	Prospective	IV	61	29	32	67 (±10)	26 (±9)	61	61	89 (±23) [60–138]	10
Haen	2015	France	Retrospective	IV	64	/	/	79.8 (±11.1) [40–95]	/	66	42	50.4	12
Simian	2015	France	Retrospective	IV	47	/	/	67.9 (±9.3) [38–90]	/	47	47	87.6 [60–137]	11
van Heumen	2015	The Netherlands	Retrospective	IV	46	/	/	67 [32–90]	27.2 [16.6–43.0]	46	46	29 [12–66]	11
Carulli C	2016	Italy	Retrospective	IV	11	13	18	75.4 [71–86]	/	11	11	45.6 [24–84]	11
Luthra	2016	Oman	Retrospective	IV	63	35	30	61 [23–91]	/	63	63	60 [18–72]	11
Plummer	2016	USA	Retrospective	IV	11	/	/	64 (42–87)	28.6 [20.8–43.6]	11	11	28.8 [24–48]	12
Hamadouche	2017	France	Retrospective	IV	51	12	39	71.4 (±11.5) [41.1–91.8]	26.4 (±6.5) [17.6–56.6]	51	30	98.4 [60–156]	10
Lebeau	2017	France	Retrospective	IV	62	20	42	70.5 (±11.1) [36–94]	27.2 (±4.8) [19.5–43]	62	62	76.8 [60–108]	10
Mohaddes	2017	Sweden	Retrospective comparative	III	436	154	282	75 (±9)	/	436	436	37.2	20
Nabil	2017	Egypt	Prospective	IV	12	/	/	56.6 [34–63]	/	12	12	24	20
Stambough	2017	USA	Retrospective	IV	8	/	/	60.6 [51–71]	/	8	8	34.8 [24–63.6]	12
Bruggemann	2018	Sweden	Retrospective comparative	III	69	34	35	67 [35–88]	/	69	44	58.8 [6–106.8]	17
Chalmers	2018	USA	Retrospective	IV	18	6	12	64 [37–81]	28 [19–47]	18	18	36 [24–60]	12
Evangelista	2018	USA	Retrospective	IV	18	9	9	62 [30–86]	29 [19–37]	18	16	36 [25–56]	11
Hipfl	2018	Germany	Retrospective	IV	15	/	/	70 [42–85]	26 [17–38]	15	15	47 [25–84]	11
Kavcic	2018	Slovenia	Retrospective	IV	174	47	127	76.8 [54–98]	/	174	156	92.4 [60–120]	11
Ozden	2018	Turkey	Prospective	IV	14	3	11	64.5 [33–89]	28.8 (±5.2) [18.7–36.2]	15	14	38.1 [24–98]	13
Rashed	2018	Egypt	Prospective	IV	31	16	15	66.4 (± 5.9)	/	32	31	12	13
Spaans	2018	The Netherlands	Retrospective comparative	III	96	38	58	73.1 (±8.5)	26.3 (4.5)	102	96	27.6 [3–84]	19
Stucinskas	2018	Lithuanian	Retrospective	IV	236	96	151	72 (±12)	/	236	236	24	10
Tabori-Jensen	2018	Denmark	Prospective comparative	III	56	10	46	76.5 (42–93)	/	56	56	36	19
Wegrzyn	2018	France	Prospective	IV	126	48	78	64 (±13)	24 (±4)	131	126	33 (±17)	12
Assi	2019	Lebanon	Retrospective comparative	III	16	3	13	69.2 (±14.8)	/	16	16	72.9 (±40.5)	20
Dikmen	2019	Turkey	Prospective	IV	30	3	27	66.1 [33–89]	26.8 (±7.2) [19.7–36.2]	30	30	42.2 [24.6–75.1]	12
Fathy	2019	Egypt	Prospective	IV	20	12	8	66.8 [55–80]	/	20	20	24	14
Gabor	2019	USA	Retrospective	IV	38	18	20	62.7 (±9.7)	29.7 (±7.0)	38	38	17.9	10
Giunta	2019	France	Retrospective	IV	27	23	4	68.5 (±8.1) [60–84]	/	27	25	48 [12–84]	12
Plummer	2019	USA	Retrospective	IV	19	/	/	64 [48–81]	27.9 [18.5–38]	19	13	24	12
Schmidt-Braekling	2019	Germany	Retrospective	IV	79	24	55	68.5 [41–87]	26.8 [18.6–41.5]	79	71	63.6 [24–122.4]	12
Wheelton	2019	England (UK)	Retrospective	IV	54	12	42	78 [49–97]	/	54	54	22.8 [6–60]	11
de l’Escalopier	2020	France	Retrospective	III	76	23	58	71 [31–90]	25.2 [17.2–38]	76	63	76.8 [36–144]	10
Lannes	2020	Switzerland	Retrospective comparative	III	26	15	11	78 (±6) [66–88]	/	26	25	12 [1–96]	19
Lavignac	2020	France	Retrospective	IV	71	27	64	62 (±10.5) [38–88]	/	71	71	28.2 (±2.9) [0.3–124.8]	10
Mahmoud	2020	Egypt	Prospective	IV	20	11	9	65.85 (±5.58) [58–78]	/	20	20	24	14
Sayac	2020	France	Retrospective	IV	74	24	50	70 (±11.3) [34–88]	/	77	39	128.4 [25.2–194.4]	12
Schmidt	2020	France	Retrospective comparative	IV	59	/	/	69 (±13.2) [19–92]	26.5 (±5.1) [17–46]	59	59	24 [12–141.6]	20
Tabori-Jensen	2020	Denmark	Prosective randomized trial	I	29	14	15	75 [70–82]	28 [23–39]	29	29	24	24
Wegrzyn	2020	Switzerland	Retrospective	IV	28	17	11	82 [74–93]	25 [20–39]	28	28	42 [24–60]	12
Bellova	2021	Germany	Retrospective	IV	33	/	/	78.6 (±7.1) [63–93]	/	33	19	28.5 (±17.3) [3–64]	10
Bozon	2021	france	Retrospective comparative	III	23	12	11	67 (±10)	24 (±3)	23	23	108 (±12)	18
Elkhadrawe	2021	Egypt	Prospective	IV	31	16	15	66.6 (±6.3)	/	32	30	12	13
Lamo Espinosa	2021	Spain	Retrospective	IV	68	15	53	81.7 (± 6.4)	/	68	68	49 (±22.6)	12
Moreta	2021	Spain	Retrospective	IV	10	/	/	79.2 [71–87]	27.5 [19–34]	10	10	42 [24–72]	12
Rashed	2021	Egypt	Prospective	I	31	16	15	66.38 [63.9–68.7]	/	31	30	12	23
Unter Ecker	2021	Germany	Retrospective	III	216	96	120	69 (±9)	29 (±7)	216	216	69 [60–110]	10
Uriarte	2021	Spain	Retrospective comparative	IV	44	13	31	76.3 (±6.7)	25.8 (±4.1)	44	44	49.2	20
**Totals and proportions**					**3426**	**1018 (37.1%)**	**1729 (62.9%)**		**Mean: 26.8 kg/m^2^**	**3452**	**3239 (94.5%)**		

**Table 2 jpm-13-00081-t002:** Cemented dual mobility cup: surgical information.

Author	Year	Cemented DMC	Posterior Approach	Lateral Approach	Anterior Approach	Extensile Approach	Avantage (Zimmer Biomet)	Polarcup (Smith and Nephew)	Ecofit (Implantcast)	Tornier	ADM (Stryker)	MDM (Stryker)	Medial Cup (Aston Medical)	SaturneTM (Amplitude)	Novae (SERF)	Symbol Cup DM (Dedienne santé)	Quattro (Lepine)	Apogee (Biotechni Inc.)	ADES (ZimmerBiomet)	DMS (SEM)	Cement Palacos R+G (Heraeus)	CMW Type 3 with Gentamycin (DePuy)	Antibiotic-Loaded Cement Simplex (Stryker)	Graft	AutoGraft	AlloGraft	Synthetic Graft
Langlais	2008	88	40	0	0	48	0	0	0	0	0	0	88	0	0	0	0	0	0	0	0	0	0	0	0	0	0
Philippot	2009	51	/	/	/	/	0	0	0	0	0	0	0	0	51	0	0	0	0	0	/	/	/	/	/	/	/
Tarasevicius	2010	42	42	0	0	0	42	0	0	0	0	0	0	0	0	0	0	0	0	0	0	0	0	0	0	0	0
Schneider	2011	96	96	0	0	0	0	0	0	0	0	0	0	0	96	0	0	0	0	0	0	0	0	91	0	0	0
Civinini	2012	24	/	/	/	/	24	0	0	0	0	0	0	0	0	/	/	/	0	0	1	0	0	30	0	30	0
Hailer	2012	200	/	/	/	/	200	0	0	0	0	0	0	0	0	0	0	0	0	0	0	0	0	0	0	0	0
Pattyn	2012	37	37	0	0	0	0	0	0	0	0	0	0	0	0	0	0	37	0	0	1	0	0	0	0	0	0
Mukka	2013	34	34	0	0	0	34	0	0	0	0	0	0	0	0	0	0	0	0	0	0	0	0	0	0	0	0
Toro-Ibarguen	2014	14	/	/	/	/	0	14	0	0	0	0	0	0	0	0	0	0	0	0	0	0	0	0	0	0	0
Wegrzyn	2014	61	61	0	0	0	0	0	0	0	0	0	0	61	0	0	0	0	0	0	1	0	0	61	0	61	0
Haen	2015	66	66	0	0	0	0	0	0	0	0	0	0	66	0	0	0	0	0	0	1	0	0	3	1	2	0
Simian	2015	47	47	0	0	0	0	0	0	0	0	0	0	0	0	0	0	0	0	47	0	0	0	23	0	23	0
van Heumen	2015	46	46	0	0	0	46	0	0	0	0	0	0	0	0	0	0	0	0	0	0	0	0	6	0	0	0
Carulli C	2016	11	0	11	0	0	11	0	0	0	0	0	0	0	0	0	0	0	0	0	/	/	/	/	/	/	/
Luthra	2016	63	65	0	0	0	63	0	0	0	0	0	0	0	0	0	0	0	0	0	1	0	0	0	0	0	0
Plummer	2016	11	10	0	0	1	0	3	0	0	6	2	0	0	0	0	0	0	0	0	0	0	0	0	0	0	0
Hamadouche	2017	51	17	3	0	31	0	0	0	0	0	0	51	0	0	0	0	0	0	0	0	1	0	15	1	13	1
Lebeau	2017	62	62	0	0	0	0	0	0	0	0	0	0	0	0	0	62	0	0	0	1	0	0	58	0	58	0
Mohaddes	2017	436	283	0	140	0	436	0	0	0	0	0	0	0	0	0	0	0	0	0	0	0	0	0	0	0	0
Nabil	2017	12	0	12	0	0	/	/	/	/	/	/	/	/	/	/	/	/	/	/	0	0	0	1	0	1	0
Stambough	2017	8	8	0	0	0	0	0	0	0	2	6	0	0	0	0	0	0	0	0	0	0	0	1	0	0	0
Bruggemann	2018	69	0	69	0	0	69	0	0	0	0	0	0	0	0	0	0	0	0	0	1	0	0	16	0	0	0
Chalmers	2018	18	11	0	7	0	0	0	0	0	18	0	0	0	0	0	0	0	0	0	0	0	0	0	0	0	0
Evangelista	2018	18	0	0	0	0	0	18	0	0	0	0	0	0	0	0	0	0	0	0	0	0	0	0	0	0	0
Hipfl	2018	15	0	0	0	0	0	15	0	0	0	0	0	0	0	0	0	0	0	0	1	0	0	15	0	15	0
Kavcic	2018	174	0	0	174	0	174	0	0	0	0	0	0	0	0	0	0	0	0	0	0	0	0	0	0	0	0
Ozden	2018	15	15	0	0	0	0	15	0	0	0	0	0	0	0	0	0	0	0	0	0	0	0	5	0	0	0
Rashed	2018	32	32	0	0	0	0	0	32	0	0	0	0	0	0	0	0	0	0	0	0	0	0	0	0	0	0
Spaans	2018	102	102	0	0	0	102	0	0	0	0	0	0	0	0	0	0	0	0	0	0	0	1	102	0	102	0
Stucinskas	2018	236	236	8	3	0	227	0	0	0	0	0	0	0	0	0	9	0	0	0	0	0	0	0	0	0	0
Tabori-Jensen	2018	56	56	0	0	0	0	0	0	0	0	0	0	56	0	0	0	0	0	0	1	0	0	0	0	0	0
Wegrzyn	2018	131	131	0	0	0	0	0	0	0	0	0	0	0	131	0	0	0	0	0	1	0	0	0	0	0	0
Assi	2019	16	16	0	0	0	/	/	/	/	/	/	/	/	/	/	/	/	/	/	0	0	0	16	0	0	0
Dikmen	2019	30	30	0	0	0	0	30	0	0	0	0	0	0	0	0	0	0	0	0	0	0	0	14	0	0	0
Fathy	2019	20	0	20	0	0	20	0	0	0	0	0	0	0	0	0	0	0	0	0	0	0	0	4	0	0	0
Gabor	2019	38	27	11	0	0	0	38	0	0	0	0	0	0	0	0	0	0	0	0	0	0	0	18	0	18	0
Giunta	2019	27	27	0	6	0	0	0	0	27	0	0	0	0	0	0	0	0	0	0	0	0	0	6	6	0	0
Plummer	2019	19	19	0	0	0	0	0	0	0	0	19	0	0	0	0	0	0	0	0	0	0	0	19	0	0	19
Schmidt-Braekling	2019	79	/	/	/	/	60	0	19	0	0	0	0	0	0	0	0	0	0	0	0	0	0	0	0	0	0
Wheelton	2019	54	0	0	0	0	/	/	/	/	/	/	/	/	/	0	0	0	0	0	0	0	0	0	0	0	0
de l’Escalopier	2020	76	0	0	0	76	0	0	0	0	0	0	76	0	0	0	0	0	0	0	0	1	0	39	0	39	0
Lannes	2020	26	26	0	0	0	0	0	0	0	0	0	0	0	0	26	0	0	0	0	0	0	0	0	0	0	0
Lavignac	2020	71	0	0	0	0	/	/	/	/	/	/	/	/	/	0	0	0	0	0	0	0	0	0	0	0	0
Mahmoud	2020	20	0	20	0	0	/	/	/	/	/	/	/	/	/	0	0	0	0	0	0	0	0	0	0	0	0
Sayac	2020	77	77	0	0	0	0	0	0	0	0	0	0	0	77	0	0	0	0	0	0	0	0	0	0	0	0
Schmidt	2020	59	59	0	0	0	0	0	0	0	0	0	0	0	0	0	59	0	0	0	0	0	0	0	0	0	0
Tabori-Jensen	2020	29	29	0	0	0	29	0	0	0	0	0	0	0	0	0	0	0	0	0	1	0	0	0	0	0	0
Wegrzyn	2020	28	28	0	0	0	0	0	0	0	0	0	0	0	0	28	0	0	0	0	1	0	0	0	0	0	0
Bellova	2021	33	33	0	0	0	0	29	4	0	0	0	0	0	0	0	0	0	0	0	1	0	0	0	0	0	0
Bozon	2021	23	23	0	0	0	0	0	0	23	0	0	0	0	0	0	0	0	0	0	0	0	0	21	0	21	0
Elkhadrawe	2021	32	32	0	0	0	/	/	/	/	/	/	/	/	/	/	/	/	/	/	0	0	0	0	0	0	0
Lamo Espinosa	2021	68	0	0	68	0	6	0	0	0	0	0	0	0	51	0	0	0	11	0	0	0	0	0	0	0	0
Moreta	2021	10	4	0	6	0	0	0	0	0	0	0	0	0	10	0	0	0	0	0	0	0	0	0	0	0	0
Rashed	2021	31	31	0	0	0	0	0	31	0	0	0	0	0	0	0	0	0	0	0	0	0	0	0	0	0	0
Unter Ecker	2021	216	216	0	0	0	216	0	0	0	0	0	0	0	0	0	0	0	0	0	0	0	0	0	0	0	0
Uriarte	2021	44	0	0	0	0	0	0	0	0	0	0	0	0	44	0	0	0	0	0	0	1	0	0	0	0	0

**Table 3 jpm-13-00081-t003:** Cemented dual mobility cup: surgical setting.

Author	Year	Cemented DMC	Primary Setting	Revision Setting	No Acetabular Reinforcement	Acetabular Reinforcement	Cup-in-Cup
Langlais	2008	88	0	88	7	81	0
Philippot	2009	51	0	51	0	51	0
Tarasevicius	2010	42	42	0	42	0	0
Schneider	2011	96	0	96	0	96	0
Civinini	2012	24	0	24	0	24	0
Hailer	2012	200	0	200	/	/	0
Pattyn	2012	37	0	37	0	37	0
Mukka	2013	34	9	25	34	0	0
Toro-Ibarguen	2014	14	0	14	0	14	0
Wegrzyn	2014	61	0	61	0	61	0
Haen	2015	66	32	34	66	0	0
Simian	2015	47	0	47	24	23	0
van Heumen	2015	46	0	46	46	0	0
Carulli C	2016	11	0	11	11	0	0
Luthra	2016	63	30	33	63	0	0
Plummer	2016	11	0	11	0	11	0
Hamadouche	2017	51	0	51	29	22	0
Lebeau	2017	62	0	62	0	62	0
Mohaddes	2017	436	0	436	436	0	0
Nabil	2017	12	12	0	11	1	0
Stambough	2017	8	0	8	0	0	8
Bruggemann	2018	69	0	69	0	69	0
Chalmers	2018	18	0	18	0	0	18
Evangelista	2018	18	0	18	0	0	18
Hipfl	2018	15	0	15	0	15	0
Kavcic	2018	174	173	1	174	0	0
Ozden	2018	15	0	15	6	9	0
Rashed	2018	32	32	0	32	0	0
Spaans	2018	102	0	102	102	0	0
Stucinskas	2018	236	0	236	236	0	0
Tabori-Jensen	2018	56	56	0	56	0	0
Wegrzyn	2018	131	131	0	0	131	0
Assi	2019	16	0	16	0	16	0
Dikmen	2019	30	0	30	11	19	0
Fathy	2019	20	0	20	20	0	0
Gabor	2019	38	0	38	27	11	0
Giunta	2019	27	27	0	4	23	0
Plummer	2019	19	19	0	0	19	0
Schmidt-Braekling	2019	79	0	79	0	79	0
Wheelton	2019	54	0	54	54	0	0
de l’Escalopier	2020	76	0	76	23	53	0
Lannes	2020	26	26	0	0	26	0
Lavignac	2020	71	71	0	0	71	0
Mahmoud	2020	20	15	5	20	0	0
Sayac	2020	77	0	77	0	77	0
Schmidt	2020	59	0	59	0	59	0
Tabori-Jensen	2020	29	29	0	29	0	0
Wegrzyn	2020	28	0	28	0	0	28
Bellova	2021	33	0	33	0	0	33
Bozon	2021	23	0	23	0	23	0
Elkhadrawe	2021	32	32	0	32	0	0
Lamo Espinosa	2021	68	68	0	68	0	0
Moreta	2021	10	0	10	0	0	10
Rashed	2021	31	31	0	31	0	0
Unter Ecker	2021	216	0	216	NA	NA	NA
Uriarte	2021	44	44	0	44	0	0
**Totals and proportions**		**3452**	**879 (25.5%)**	**2573 (74.5%)**	**1738 (57.2%)**	**1183 (39%)**	**115 (3.8%)**

**Table 4 jpm-13-00081-t004:** Cemented dual mobility cup: complications.

Author	Year	C-DMC	Dislocations	Intra-Prosthetic Dislocations	Infection	Aseptic Loosening	Mechanical Failure	Revisions	Cup-Survivorship
Langlais	2008	88	0	1	2	0	0	3	100%
Philippot	2009	51	6	0	3	2	1	/	86.8%
Tarasevicius	2010	42	0	0	0	0	0	0	87%
Schneider	2011	96	11	0	5	1	2	4	96%
Civinini	2012	24	0	0	1	0	0	1	100%
Hailer	2012	200	3	0	4	4	2	10	/
Pattyn	2012	37	2	0	1	0	1	2	97%
Mukka	2013	34	2	0	3	0	0	0	91.2%
Toro-Ibarguen	2014	14	1	0	/	/	/	/	91.2%
Wegrzyn	2014	61	0	0	0	0	1	0	96%
Haen	2015	66	0	0	0	1	0	1	100%
Simian	2015	47	1	0	6	0	0	/	100%
van Heumen	2015	46	0	0	2	1	0	3	100%
Carulli C	2016	11	0	0	0	0	0	0	/
Luthra	2016	63	1	0	1	0	0	1	/
Plummer	2016	11	1	0	2	0	2	4	98%
Hamadouche	2017	51	1	2	2	2	0	7	93%
Lebeau	2017	62	1	0	7	5	0	8	75.2%
Mohaddes	2017	436	7	0	15	2	2	10	89%
Nabil	2017	12	0	0	0	0	0	0	100%
Stambough	2017	8	2	0	0	0	0	/	98.5%
Bruggemann	2018	69	1	0	0	2	0	2	94.6%
Chalmers	2018	18	2	1	0	0	0	2	92.3%
Evangelista	2018	18	0	0	0	0	0	0	/
Hipfl	2018	15	1	0	2	0	0	/	91.9%
Kavcic	2018	174	0	0	2	0	0	0	98%
Ozden	2018	15	0	0	0	2	0	2	100%
Rashed	2018	32	0	0	1	0	0	1	96%
Spaans	2018	102	3	0	1	4	0	5	/
Stucinskas	2018	236	5	0	5	1	0	11	/
Tabori-Jensen	2018	56	1	0	0	1	0	/	100%
Wegrzyn	2018	131	3	0	4	0	0	0	93%
Assi	2019	16	0	0	0	0	0	0	98.80%
Dikmen	2019	30	1	0	1	2	0	3	/
Fathy	2019	20	1	0	0	0	0	0	96.80%
Gabor	2019	38	1	0	1	0	0	1	93.75
Giunta	2019	27	3	0	0	0	0	0	/
Plummer	2019	19	0	0	2	0	0	1	72.1%
Schmidt-Braekling	2019	79	2	0	5	4	0	9	90%
Wheelton	2019	54	0	0	1	0	0	1	95.8%
de l’Escalopier	2020	76	2	3	2	1	0	/	/
Lannes	2020	26	2	0	2	0	0	2	85%
Lavignac	2020	71	1	0	10	4	6	/	83.10%
Mahmoud	2020	20	0	0	0	0	0	0	100%
Sayac	2020	77	7	0	2	3	0	6	85%
Schmidt	2020	59	4	0	/	/	/	13	95.6%
Tabori-Jensen	2020	29	0	0	0	0	0	0	100%
Wegrzyn	2020	28	0	0	0	0	0	0	/
Bellova	2021	33	2	0	1	0	1	2	97.3%
Bozon	2021	23	1	0	3	2	0	2	93%
Elkhadrawe	2021	32	0	0	1	0	0	1	98%
Lamo Espinosa	2021	68	0	0	0	1	0	1	/
Moreta	2021	10	1	0	0	0	0	1	92.2%
Rashed	2021	31	0	0	1	0	0	0	/
Unter Ecker	2021	216	24	0	0	0	0	17	100%
Uriarte	2021	44	0	0	2	0	0	1	/
**Totals and proportions**		**3452**	**107 (3.1%)**	**7 (0.2%)**	**103 (3%)**	**45 (1.3%)**	**18 (0.5%)**	**138 (4%)**	**Mean: 93.5%**

**Table 5 jpm-13-00081-t005:** Cemented dual mobility cup: radiographic and functional outcomes.

Author	Patiens	Patients at Follow-Up	Acetabular Components Radiolucent Lines	Brooker Heterotopic Ossification	Pre-Operative HHS	Post-Operative HHS	Pre-Operative PMA	Post-Operative PMA
Langlais	82	79	2	/	26.2	87.4 ± 12.1	/	/
Philippot	51	51	/	/	/	59.4 ± 22.2 (29–91)	/	/
Tarasevicius	42	42	3	/	/	82 ± 18 (40–100)	11 ± 3	15.5 ± 3 (11–18)
Schneider	96	77	0	/	/	77 (25–100)	/	/
Civinini	24	24	3	4	/	/	/	/
Hailer	200	200	0	/	47 (37–60)	81 (62–98)	/	/
Pattyn	36	36	15	/	48 (32–68)	86 (66–95)	/	/
Mukka	34	34	0	/	/	/	12.9 (5–18)	16.3 (10–18)
Toro-Ibarguen	14	14	/	/	42.8 ± 6.7 (34–60)	87.3 ± 5.8 (75–98)	/	/
Wegrzyn	61	61	/	/	/	/	/	/
Haen	64	42	0	2	/	92.6 ± 11.1	/	/
Simian	47	51	/	/	46 (40–79)	65 (41–97)	/	/
van Heumen	46	46	1	/	40 (23–44)	86 (79–96)	/	/
Carulli C	11	11	0	/	50 (35–78)	78 (49–95)	/	/
Luthra	63	63	/	/	/	70.4 ± 23 (24–90)	/	14.3 ± 4.2 (7–18)
Plummer	11	11	3	0	/	/	/	15.5 ± 1.9 (9–18)
Hamadouche	51	30	0	0	/	/	/	/
Lebeau	62	62	9	/	/	/	13.5 ± 4.0 (8–18)	16.3 ± 1.6 (13–18)
Mohaddes	436	436	0	/	30 (15–51)	71 (40–89)	/	/
Nabil	12	12	0	0	31.7 (20–81)	84.5 (32–100)	/	/
Stambough	8	12	/	/	/	/	10.31	15.61
Bruggemann	69	44	3	/	/	/	/	16.1
Chalmers	18	18	0	6	/	72.36 ± 11.65	/	/
Evangelista	18	16	/	/	/	/	/	/
Hipfl	15	15	2	/	49 ± 16 (17–90)	73 ± 21 (24–99)	11 ± 3 (3–18)	14.4 ± 3.6 (3–18)
Kavcic	174	156	0	0	/	/	/	/
Ozden	14	14	/	/	25.95 ± 9.91 (14–44)	92.45 ± 2.74 (88–98)	/	/
Rashed	31	31	/	/	/	/	/	/
Spaans	96	96	0	0	49.3 (33–62)	71.3 (22–91)	10.1	12.8
Stucinskas	236	236	/	/	/	67 (±14)	/	/
Tabori-Jensen	56	56	/	0	39.4	87.6	/	/
Wegrzyn	126	25	0	0	42	86 (49–93)	/	/
Assi	16	16	/	/	/	/	/	/
de l’Escalopier	76	63	0	0	45	90	/	/
Dikmen	30	30	0	0	34 (27–41)	82 (66–95)	/	/
Fathy	20	20	2	0	/	92.8 (88.2–97.4)	/	/
Gabor	38	38	0	2	/	92.8 (SD 11.1)	/	/
Giunta	27	25	0	0	39.95 (6–84)	/	8.05 (3–16)	/
Moreta	10	10	4	29	39.5 ± 9.6 [37–43]	71.3 ± 14	8.1 ± 2.5 [7–9]	15.3 ± 2.2 [15–16]
Plummer	19	13	/	/	/	/	/	/
Sayac	74	39	/	/	/	73.0 (24–99)	/	/
Schmidt	59	59	6	17	/	/	9.6 ± 3.06 (2–16)	15.5 ± 2.32 (7–18)
Schmidt-Braekling	79	71	3	0	/	80.4 ± 12.9 (51–98)	/	15.2 ± 2 (11–18)
Wheelton	54	54	/	/	/	/	/	/
Lannes	26	25	/	/	/	/	/	/
Lavignac	71	91	/	/	/	/	/	/
Mahmoud	20	20	0	0	56 (±12)	92 (8.7)	/	/
Rashed	31	30	9	7	/	78.8 (31–100)	/	/
Tabori-Jensen	29	29	/	/	/	/	/	/
Wegrzyn	28	28	/	/	/	/	5.48 (SD 2.41)	10.5 (SD 3.82)
Bellova	33	19	4	16	/	76.9 (16.8)	/	13.1 (3.3)
Bozon	23	23	0	0	/	/	/	/
Elkhadrawe	31	30	3	0	53 ± 19	79 ± 13	/	/
Lamo Espinosa	68	68	0	0	37 ± 8	84 ± 7	/	/
Unter Ecker	216	216	0	0	71 (69–74)	88 (82–95)	/	/
Uriarte	44	44	/	0	/	/	/	/
**Totals and proportions**	**3426**	**3162 (92.3%)**	**72 (3.2%)**	**83 (6.7%)**	**Mean: 43**	**Mean: 76.7**	**Mean: 10.4**	**Mean: 14.7**

**Table 6 jpm-13-00081-t006:** Cemented dual mobility cup in the primary setting.

Author	Year	C-DMC Primary Setting	Primary (FNF)	Primary (AO)	Primary (Oncology)	Primary (Acetabular Fracture)	No Acetabular Reinforcement	Acetabular Reinforcement	Dislocations	Intra-Prosthetic Dislocations	Revisions	Mechanical Failure	Aseptic Loosening	Infection	Posterior Approach	Lateral Approach	Anterior Approach	Cup-Survivorship
Tarasevicius	2010	42	42	0	0	0	42	0	0	0	0	0	0	0	42	0	0	100%
Mukka	2013	9	0	9	0	0	9	0	2	0	0	0	0	3	9	0	0	94.1%
Haen	2015	32	12	20	0	0	32	0	0	0	1	0	1	0	32	0	0	98% [94–100]
Luthra	2016	30	9	18	0	3	30	0	1	0	1	0	0	1	30	0	0	98%
Nabil	2017	12	0	12	0	0	11	1	0	0	0	0	0	0	0	12	0	100%
Kavcic	2018	173	88	85	0	0	173	0	0	0	0	0	0	2	0	0	173	100%
Rashed	2018	32	32	0	0	0	32	0	0	0	1	0	0	1	32	0	0	93.75%
Tabori-Jensen	2018	56	56	0	0	0	56	0	/	0	/	0	1	0	56	0	0	/
Wegrzyn	2018	131	0	0	131	0	0	131	3	0	0	0	0	4	131	0	0	/
Giunta	2019	27	0	0	0	27	4	23	3	0	0	0	0	0	27	0	6	/
Plummer	2019	19	0	0	19	0	0	19	0	0	1	0	0	2	19	0	0	/
Lannes	2020	26	0	0	0	26	0	26	2	0	2	0	0	2	26	0	0	92.3%
Lavignac	2020	71	0	0	71	0	0	71	1	0	/	6	4	10	/	/	/	/
Mahmoud	2020	15	10	4	1	0	15	0	0	0	0	0	0	0	0	15	0	100%
Tabori-Jensen	2020	29	0	29	0	0	29	0	0	0	0	0	0	0	29	0	0	100%
Elkhadrawe	2021	32	32	0	0	0	32	0	0	0	1	0	0	1	32	0	0	100%
Lamo Espinosa	2021	68	0	68	0	0	68	0	0	0	1	0	1	0	0	0	68	98.5%
Rashed	2021	31	31	0	0	0	31	0	0	0	0	0	0	1	31	0	0	/
Uriarte	2021	44	44	0	0	0	44	0	0	0	1	0	0	2	/	/	/	97.3% [93.5–100]
**Totals and proportions**		**879**	**356 (40.5%)**	**245 (27.9%)**	**222 (25.2%)**	**56 (6.4%)**	**608 (69.2%)**	**271 (30.8%)**	**12 (1.5%)**	**0**	**9 (1.2%)**	**6 (0.7%)**	**7 (0.8%)**	**29 (8.1%)**	**496 (64.4%)**	**27 (3.5%)**	**247 (32.1%)**	**Mean: 98.5 %**

**Table 7 jpm-13-00081-t007:** Cemented dual mobility cup in the revision setting.

Author	Year	C-DMC Revision Setting	No Acetabular Reinforcement	Acetabular Reinforcement	Cup-in-Cup	Posterior/Posterolateral Approach	Lateral Approach	Anterior/Anterolateral Approach	Extensile Approach	Graft	Complications (Dislocations)	Complications (IPD)	Complications (Revisions)	Complications (Mechanical Failure)	Complications (Aseptic Loosening)	Complications (Infection)	Cup-Survivorship
Langlais	2008	88	7	81	0	40	0	0	48	0	0	1	3	0	0	2	94.6%
Philippot	2009	51	0	51	0	/	/	/	/	/	6	0	/	1	2	3	98.8%
Schneider	2011	96	0	96	0	96	0	0	0	91	11	0	4	2	1	5	95.6% (95% CI, 93.3–97.7%)
Civinini	2012	24	0	24	0	/	/	/	/	30	0	0	1	0	0	1	97% (95% CI, 82–98%)
Hailer	2012	200	/	/	0	/	/	/	/	0	3	0	10	2	4	4	93% (95% CI, 90–97%)
Pattyn	2012	37	0	37	0	37	0	0	0	0	2	0	2	1	0	1	/
Mukka	2013	25	25	0	0	25	0	0	0	0	2	0	0	0	0	3	94.11%
Toro-Ibarguen	2014	14	0	14	0	/	/	/	/	0	1	0	/	/	/	/	/
Wegrzyn	2014	61	0	61	0	61	0	0	0	61	0	0	0	1	0	0	98%
Haen	2015	34	34	0	0	34	0	0	0	3	0	0	1	0	1	0	98% (95% CI, 94–100%)
Simian	2015	47	24	23	0	47	0	0	0	23	1	0	/	0	0	6	90% (95% CI, 84–95%)
van Heumen	2015	46	46	0	0	46	0	0	0	6	0	0	3	0	1	2	93% (95% CI, 79–98%)
Carulli C	2016	11	11	0	0	0	11	0	0	/	0	0	0	0	0	0	100%
Luthra	2016	33	33	0	0	33	0	0	0	0	1	0	1	0	0	1	98%
Plummer	2016	11	0	11	0	10	0	0	1	0	1	0	4	2	0	2	/
Hamadouche	2017	51	29	22	0	17	3	/	31	15	1	2	7	0	2	2	75.2 ± 9.3% (95% CI, 56.9–93.5%)
Lebeau	2017	62	0	62	0	62	0	0	0	58	1	0	8	0	5	7	91.9%
Mohaddes	2017	436	436	0	0	285	/	140	/	0	7	0	10	2	2	15	96%
Stambough	2017	8	0	0	8	8	0	0	0	1	2	0	/	0	0	0	85%
Bruggemann	2018	69	0	69	0	0	69	0	0	16	1	0	2	0	2	0	96% (95% CI, 90–100%)
Chalmers	2018	18	0	0	18	11	0	7	0	0	2	1	2	0	0	0	/
Evangelista	2018	18	0	0	18	/	/	/	/	0	0	0	0	0	0	0	100%
Hipfl	2018	15	0	15	0	/	/	/	/	15	1	0	/	0	0	2	89 (72–96)
Ozden	2018	15	6	9	0	15	0	0	0	5	0	0	2	0	2	0	93% (95% CI, 88–98.7%)
Spaans	2018	102	102	0	0	102	0	0	0	102	3	0	5	0	4	1	95.8% (3 months–7 years) (95% CI, 91.7–99.9%)
Stucinskas	2018	236	236	0	0	/	/	/	/	0	5	0	11	0	1	5	95.14%
Assi	2019	16	0	16	0	16	0	0	0	16	0	0	0	0	0	0	100%
de l’Escalopier	2019	76	23	53	0	0	0	0	76	39	2	3	/	0	1	2	91.2 ± 3.8%
Dikmen	2019	30	11	19	0	30	0	0	0	14	1	0	3	0	2	1	91.2% (95% CI, 81.6–100%)
Fathy	2019	20	20	0	0	0	20	0	0	4	1	0	0	0	0	0	100%
Gabor	2019	38	27	11	0	27	11	0	0	18	1	0	1	0	0	1	/
Moreta	2019	10	0	0	10	4	0	6	0	0	1	0	1	0	0	0	/
Sayac	2019	77	0	77	0	77	0	0	0	0	7	0	6	0	3	2	92.2%
Schmidt	2019	59	0	59	0	59	0	0	0	0	4	0	13	/	/	/	72.1%
Schmidt-Braekling	2019	79	0	79	0	/	/	/	/	0	2	0	9	0	4	5	85%
Wheelton	2019	54	54	0	0	/	/	/	/	0	0	0	1	0	0	1	/
Mahmoud	2020	5	5	0	0	0	5	0	0	0	0	0	0	0	0	0	100%
Wegrzyn	2020	28	0	0	28	28	0	0	0	0	0	0	0	0	0	0	100%
Bellova	2021	33	0	0	33	33	0	0	0	0	2	0	2	1	0	1	86.8%
Bozon	2021	23	0	23	0	23	0	0	0	21	1	0	2	0	2	3	87% (95% CI, 94.7–72.3)
Unter Ecker	2021	216	/	/	/	216	0	0	0	0	24	0	17	0	0	0	96%
**Totals and proportions**		**2572**	**1129 (52.4%)**	**912 (42.3%)**	**115 (5.3%)**	**1442 (77.1%)**	**119 (6.4%)**	**153 (8.2%)**	**156 (8.3%)**	**538 (21.4%)**	**97 (3.8%)**	**7 (0.3%)**	**131 (5.1%)**	**12 (0.5%)**	**39 (1.5%)**	**78 (3%)**	**93.6%**

**Table 8 jpm-13-00081-t008:** Cemented DMC without acetabular reinforcement.

Author	Year	C-DMC without Acetabular Reinforcement	Primary Setting	Revision Setting	Graft	Dislocations	IPD	Mechanical Failure	Aseptic Loosening	Infection	Revisions	Cup-Survivorship
Tarasevicius	2010	42	42	0	0	0	0	0	0	0	0	100%
Mukka	2013	34	9	25	0	2	0	0	0	3	3	94.1%
Carulli C	2015	11	0	11	/	0	0	0	0	0	0	100%
Haen	2015	66	32	34	3	0	0	0	1	0	1	98%
Simian	2015	24	0	24	/	/	0	0	0	/	/	/
van Heumen	2015	46	0	46	6	0	0	0	1	2	3	93%
Luthra	2016	63	30	33	0	0	0	0	0	1	1	98%
Hamadouche	2017	29	0	29	/	/	/	0	/	/	/	/
Mohaddes	2017	436	0	436	0	7	0	2	2	15	10	96%
Nabil	2017	11	11	0	1	0	0	0	0	0	0	100%
Kavcic	2018	174	173	1	0	0	0	0	0	2	0	100%
Ozden	2018	6	0	6	/	0	0	0	/	0	/	/
Rashed	2018	32	32	0	0	0	0	0	0	1	1	93.75%
Spaans	2018	102	0	102	102	3	0	0	4	1	5	95.8%
Stucinskas	2018	236	0	236	0	5	0	0	1	5	11	94.9%
Tabori-Jensen	2018	56	56	0	0	1	0	0	1	0	/	/
de l’Escalopier	2019	23	0	23	/	/	/	0	/	/	/	/
Dikmen	2019	11	0	11	/	/	0	0	/	/	/	/
Fathy	2019	20	0	20	4	1	0	0	0	0	0	100%
Gabor	2019	27	0	27	/	/	0	0	0	/	/	/
Wheelton	2019	54	0	54	0	0	0	0	0	1	1	100%
Mahmoud	2020	20	15	5	0	0	0	0	0	0	0	100%
Rashed	2020	31	31	0	0	0	0	0	0	1	0	96.8%
Tabori-Jensen	2020	29	29	0	0	0	0	0	0	0	0	100%
Elkhadrawe	2021	32	32	0	0	0	0	0	0	1	1	100%
Lamo Espinosa	2021	68	68	0	0	0	0	0	1	0	1	98.5%
Uriarte	2021	44	44	0	0	0	0	0	0	2	1	93.3%
**Totals and proportions**		**1727**	**604 (35%)**	**1123 (65%)**	**116 (7.3%)**	**19 (1.2%)**	**0**	**2 (0.1%)**	**11 (0.7%)**	**35 (2.2%)**	**39 (2.5%)**	**Mean: 96.9%**

**Table 9 jpm-13-00081-t009:** Cemented DMC with acetabular reinforcement.

Author	Year	C-DMC with Acetabular Reinforcement	Acetabular Reinforcement	Primary Setting	Revision Setting	Graft	Dislocations	Intra-Prosthetic Dislocations	Mechanical Failure	Aseptic Loosening	Infection	Revisions	Cup-Survivorship
Langlais	2008	81	Kerboull Cross-Plate	0	81	0	0	1	0	0	2	3	94.6%
Philippot	2009	51	7 Novae Arm cage/Kerboull cross (44)	0	51	/	6	0	1	2	3	/	98.80%
Schneider	2011	96	70 Kerboull cross-plate, 6 Burch-Schneider antiprotrusio cage, 20 custom-fit Novae ARM cage	0	96	91	11	0	2	1	5	4	95.6%
Civinini	2012	24	Contour acetabular ring (Smith & Nephew, London, UK)	0	24	30	0	0	0	0	1	1	97%
Pattyn	2012	37	35 Ganz ring (Zimmer Inc., Warsaw, IN, USA), 2 Burch Schneider ring (Zimmer Inc., Warsaw, IN, USA)	0	37	0	2	0	1	0	1	2	/
Toro-Ibarguen	2014	14	15 Protrusio cage [DePuy Orthopaedics, Inc, Warsaw, IN], 22 Contour [Smith and Nephew Richards, Memphis, TN, USA]	0	14	0	1	0	/	/	/	/	/
Wegrzyn	2014	61	Kerboull Cross-Plate	0	61	61	0	0	1	0	0	0	98%
Simian	2015	23	Ganz Reinforcement Ring (Zimmer, Warsaw, IN, USA)	0	23	/	/	0	0	0	/	/	/
Plummer	2016	11	/	0	11	0	1	0	2	0	2	4	/
Hamadouche	2017	22	Kerboull acetabular reinforcement	0	22	/	/	/	0	/	/	/	/
Lebeau	2017	62	47 Müller ring, 8 Burch-Schneider ring, 4 Link reinforcement	0	62	58	1	0	0	5	7	8	91.9%
Bruggemann	2018	69	/	0	69	16	1	0	0	2	0	2	96%
Hipfl	2018	15	titanium acetabular cage (Zimmer Biomet)	0	15	15	2	0	0	0	2	/	89%
Ozden	2018	9	Contour Acetabular Reinforcement Ring (Smith & Nephew)	0	9	/	0	0	0	/	0	/	/
Wegrzyn	2018	131	Kerboull cross-plate or Burch-Schneider anti-protrusio cage	131	0	0	3	0	0	0	4	0	/
Assi	2019	16	Kerboull cross-plate	0	16	16	0	0	0	0	0	0	100%
Dikmen	2019	19	Contour Acetabular Reconstruction Ring (Smith & Nephew)	0	19	/	/	0	0	/	/	/	/
Gabor	2019	11	/	0	11	/	/	0	0	0	/	/	/
Giunta	2019	23	Kerboull cross-plate	23	0	6	3	0	0	0	0	0	/
Plummer	2019	19	triflange titanium acetabular cage (Restoration GAP Acetabular Cup; Stryker)	19	0	19	0	0	0	0	2	1	/
Schmidt-Braekling	2019	79	Burch–Schneider Cage (Zimmer, Warsaw, IN, USA)	0	79	0	2	0	0	4	5	9	85%
de l’Escalopier	2020	53	Kerboull acetabular reinforcement device (KARD, Zimmer-Biomet, Warsaw, IN, USA)	0	53	/	/	/	0	/	/	/	/
Lannes	2020	26	Ganz ring (Zimmer-Biomet^®^, Warsaw, IN, USA)	26	0	0	2	0	0	0	2	2	92.3%
Lavignac	2020	71	Kerboull cross-plate, Muller ring, Burch-Schneider cage	71	0	0	1	0	6	4	10	/	/
Sayac	2020	77	Kerboull cross-plate, Burch-Schneider antiprotrusio cage, custom-fit Novae ARM cage	0	77	0	7	0	0	3	2	6	92.2%
Schmidt	2020	59	Kerboull cross-plate, Burch-Schneider ring, or jumbo metal-back	0	59	0	4	0	/	/	/	13	72.1%
Bozon	2021	23	Kerboull reinforcement device (316 L, Aston Medical, Saint-Étienne, France)	0	23	21	1	0	0	2	3	2	87%
**Totals and proportions**		**1182**		**270 (22.8%)**	**912 (77.2%)**	**333 (33.5%)**	**48 (4.5%)**	**1 (0.09%)**	**13 (1.2%)**	**23 (2.3%)**	**51 (5.2%)**	**57 (6.4%)**	**Mean: 91.7%**

**Table 10 jpm-13-00081-t010:** Cemented DMC: cup-in-cup technique.

Author	Year	Cemented DMC	Dislocations	IPD	Revisions	Mechanical Failure	Aseptic Loosening	Infections	Acetabular Components RLLs	Brooker HO	Preoperative HHS	Postperative HHS	Preoperative PMA	Postoperative PMA	Revisions	Posterior Approach	Lateral Approach	Anterior Approach	Follow-up (Months) (SD) [Range]	Cup-Survivorship
Stambough	2017	8	2	0	2	0	0	1	/		/	/	/	/	8	8	0	0	34.8 [24–63.6]	85%
Chalmers	2018	18	2	1	2	0	0	0	0	/	47 [37–60]	81 [62-98]	/	/	18	11	0	7	36 [24–60]	/
Evangelista	2018	18	0	0	0	0	0	0	/	/	46 [40–79]	65 [41–97]	/	/	18	0	0	0	36 [25–56]	100%
Wegrzyn	2020	28	0	0	0	0	0	0	0	0	71 [69–74]	88 [82–95]	/	/	28	28	0	0	42 [24–60]	100%
Bellova	2021	33	2	0	2	1	0	1	/	/	/	59.4 (±22.2) [29–91]	/	/	33	33	0	0	28.5 (±17.3) [3–64]	86.8%
Moreta	2021	10	1	0	1	0	0	0	0	0	49.3 [33–62]	71.3 [22–91]	10.1	12.8	10	4	0	6	42 [24–72]	/
**Totals and proportions**		**115**	**4.7%**	**0.9%**	**4.7%**	**0.9%**	**0%**	**0.9%**	**0%**	**0%**	**Mean: 56.4**	**Mean: 74.8**	**Mean: 10.1**	**Mean: 12.8**	**100%**	**86.6%**	**0%**	**13.4%**	**Mean: 35.7**	**Mean: 93.6%**

## Data Availability

The authors confirm that the data supporting the findings of this study are available within the article.

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
