# Peer review of "Is Cemented Dual-Mobility Cup a Reliable Option in Primary and Revision Total Hip Arthroplasty: A Systematic Review"

_jpm, 2022, doi:10.3390/jpm13010081_

Round 1

Reviewer 1 Report

Lines 113-115: Could please analyse the methodological quality of each  study design. For randomized controlled trials (RCTs), the Cochrane Risk of Bias Tool. For observational comparative (cohort) studies, the Newcastle-Ottawa Scale. For retrospective comparative studies, the revised and validated version of the Methodological Index for Non-randomized Studies (MINORS criteria).

Line 118. Table 1 is not clear.

Line 120: Is necessary to mention the number of cementless hip and cemented hip with C-DMC? 

Lines 130-133: Could you please explain the most favorable approach for primary and revision C-DMC.

Lines 142-144: Is there a significant distinction between primary and revision regarding the survival rate?What type (implants) of acetabular reinforcement 

Lines 177- 178: Is significant to mention the type of implants regarding the acetabular reinforcement?

Line 280: Is significant to mention other complication?

Reviewer 2 Report

Ciolli et al. present a systematic review concerning the C-DMC and a specific analysis of its application. The authors do a nice job presenting the background and the overall objective of this study, while they also describe in a simple way how they worked on this review and what they found. This reviewer is quite positive. There are, however, some points that need to be altered which are presented in more detail below. 

line 59: there is a no needed big gap between "..2021." & "The following"

figure 1 (and all figures and tables): overall image quality is poor. Many letters seem faded for example on the up right box "Records removed" looks faded and with low image quality. Also the title of this figure "Flowchart" should have more details that will describe the specific figure in one or few sentences. (This is done in Table 1 correctly)

Table 1: image quality is very poor. It's difficult to read what's written. That's the case for all tables in the manuscript so they need to be changed accordingly so that the reader won't experience any difficulty.

comment for figure size: I think the size of the figures is relatively big for each page while also the quality is not good. It would be nice if the authors find a way to make them simpler while maintaining good image quality. 

line 174: a comment had been left there that needs to be removed

comment for the conclusions section: I believe it would be a nice addition to this review if the authors could elaborate further on the conclusions section. In that way the overall result and outcome of this review will be stronger. 
